# Biomarkers Suggesting Favorable Prognostic Outcomes in Sudden Sensorineural Hearing Loss

**DOI:** 10.3390/ijms21197248

**Published:** 2020-09-30

**Authors:** Jeon Gang Doo, Dokyoung Kim, Yong Kim, Myung Chul Yoo, Sung Su Kim, Jeewon Ryu, Seung Geun Yeo

**Affiliations:** 1Department of Otorhinolaryngology-Head and Neck Surgery, School of Medicine, Kyung Hee University, 23 Kyungheedae-ro, Dongdaemun-gu, Seoul 02447, Korea; riversoul@naver.com (J.G.D.); jeewon@bu.edu (J.R.); 2Department of Anatomy and Neurobiology, College of Medicine, Kyung Hee University, Seoul 02447, Korea; dkim@khu.ac.kr; 3Department of Physical Medicine & Rehabilitation, School of Medicine, Kyung Hee University, Seoul 02447, Korea; okkeun@nate.com (Y.K.); famousir@naver.com (M.C.Y.); 4Medical Research Center for Bioreaction to Reactive Oxygen Species and Biomedical Science Institute, School of Medicine, Graduate School, Kyung Hee University, Seoul 02447, Korea; sgskim@khu.ac.kr

**Keywords:** sudden sensorineural hearing loss, biomarkers, prognosis, recovery

## Abstract

Sudden sensorineural hearing loss (SSNHL) is a medical emergency, making detailed examination to determine possible causes and early treatment important. However, etiological examinations in SSNHL do not always reveal a cause, and several factors have been found to affect treatment outcomes. Various studies are being performed to determine the prognosis and effects of treatment in patients who experience sudden hearing loss, and to identify biomarkers associated with this condition. Embase, PubMed, and the Cochrane database were searched using the key words SSNHL, prognostic, and biomarker. This search identified 4 articles in Embase, 28 articles in PubMed, and 36 in the Cochrane database. Of these 68 articles, 3 were duplicates and 37 were unrelated to the research topic. After excluding these articles, the remaining 28 articles were reviewed. Factors associated with SSNHL were divided into six categories: metabolic, hemostatic, inflammatory, immunologic, oxidative, and other factors. The associations between these factors with the occurrence of SSNHL and with patient prognosis were analyzed. Low monocyte counts, low neutrophil/lymphocyte ratio (NLR) and monocyte/high-density lipoproteins (HDL) cholesterol ratio (MHR), and low concentrations of fibrinogen, platelet glycoprotein (GP) IIIa, and TNF-α were found to be associated with good prognosis. However, these factors alone could not completely determine the onset of and recovery from SSNHL, suggesting the need for future basic and clinical studies.

## 1. Introduction

Sudden sensorineural hearing loss (SSNHL) is defined as sensorineural hearing loss of more than 30 dB at more than three consecutive wavelengths occurring within 3 days. Its annual reported prevalence rates are 5–20 per 100,000 in the United States and >10 per 100,000 in Korea [1]. Its actual prevalence may be much higher, because many patients who experience natural cure within a few days do not seek medical assistance. Prevalence rates do not differ by sex, side (left or right), season, or geographic region. SSNHL can occur at any age, but it is most prevalent in individuals aged in their 30 s to 50 s. In most patients, SSNHL is idiopathic and occurs unilaterally, but 4–17% of patients have bilateral SSNHL [2,3]. This condition is likely multifactorial, as responses to treatment and prognosis vary. Viral infection and vascular dysfunction are thought to be the main causes, while other causes include bacterial infection, cochlear rupture, endolymphatic hydrops, trauma, congenital anomalies, autoimmune diseases, tumors, and ototoxicity. Early detection and treatment greatly affect prognosis, making rapid diagnosis important [1,4,5]. 

Clinical trials of drugs to treat SSNHL are difficult to perform, because of the difficulty determining control groups and because many factors affect patient prognosis. Clinical trials and research studies are attempting to identify biomarkers to determine the prognosis of patients with SSNHL and the effects of therapeutic agents. Biomarkers can include changes in DNA, RNA, proteins, or metabolites that distinguish between normal and pathological states or objectively measure responses to treatment. To serve as an accurate biomarker, the analyte of interest should play roles in normal biological processes and should indicate the progression of disease and responses to treatment. Biomarkers can be categorized according to their purposes, including those that screen for disease, indicate prognosis markers, indicate the efficacy of treatment, and determine toxicity [6,7]. Genetic biomarkers are the most actively researched target biomarkers, with cancer being the most researched target disease. Sensitive and specific markers for the early diagnosis and treatment of cancer patients are of special interest. Biomarkers in medicine are the main pillar of a paradigm shift, from the traditional treatment system based on collective diagnostic tests, experience and/or statistics, to a personalized medical system based on individual diagnostic tests. The development of more specialized biomarkers has increased their research and development costs. At present, 55% of biomarkers are for cancer-related diseases, 20% for cardiovascular diseases, 16% for neurological diseases, 7% for autoimmune diseases, and 2% for infectious diseases [8,9,10]. SSNHL is mostly idiopathic because the etiology of SSNHL is not clear, and recovery is difficult to predict. The purpose of the present study was therefore to evaluate the course of SSNHL and to identify a prognostic biomarker that can be used to predict treatment outcome. 

## 2. Biomarkers in SSNHL

The PubMed and other databases were searched for studies investigating biomarkers in SSNHL, including metabolic, hemostatic, inflammatory, immunologic, and oxidative factors. 

### 2.1. Metabolic Parameters


High-density lipoproteins(HDL), Low-density lipoproteins(LDL), total cholesterol, triglycerides (TGs)Apolipoprotein (Apo)-1, apoB, apoEAtherogenic index of plasma (AIP), atherogenic (ATH) indexLipoprotein (Lp) αMonocyte/HDL ratio (MHR)Glucose, glycated hemoglobin (HbA1c)


High-density lipoproteins (HDL) guard endothelial cells against the harmful effects of low-density lipoproteins (LDL) and prevent oxidation by LDL molecules. Therefore, HDL is known to have anti-inflammatory and anti-oxidative effects [11,12]. A study investigating biomarkers in patients with SSNHL found that age, sex, monocyte count, HDL, and MHR (monocyte/HDL ratio) did not differ between the control and SSNHL groups. After drug treatment of patients with SSNHL, however, average monocyte count (*p* = 0.002) and MHR (*p* = 0.005) were significantly lower in responders than in non-responders, suggesting that MHR may be a prognostic marker in patients with SSNHL [13]. HDL was found to be significantly lower and LDL significantly higher in patients with successive bilateral SSNHL than in patients with unilateral SSNHL. Because these factors affect blood supply, they may predict successive bilateral SSNHL [14].

Another study compared eight types of serologic, viral, and autoimmune markers, 16 types of biochemical markers, prothrombin, and factor V mutations, and five types of coagulation factors in patients with SSNHL and a control group. Compared with the control group, patients with SSNHL had higher concentrations of blood glucose, homocysteine, factor VIII, glycated hemoglobin (HbA1c), total cholesterol, LDL, and lipoprotein α (Lp α). Glucose, HbA1c, factor VIII, uric acid, and homocysteine concentrations were found to be higher in patients with more severe SSNHL, as determined by pure tone audiometry, with homocysteine and Lp α concentrations being especially higher in patients with profound hearing loss. These metabolic parameters suggest associations among SSNHL, severity of hearing loss, vascular damage and risk of thrombophilia [15].

Although total cholesterol, triglycerides (TGs), and HDL concentrations did not differ significantly between SSNHL patients and a control group, atherogenic (ATH) index, as determined by binary logistic regression analysis, was more than four-fold higher in the SSNHL group (odds ratio (OR): 4.25, 95% confidence interval (CI): 1.32–13.7, *p* = 0.013), suggesting that ATH index is a risk factor for SSNHL [16].
ATH index = ((TC − HDL-C) × apoB)/(HDL-C × apoA-I)
TC: total cholesterol; HDL-C: high-density lipoprotein cholesterol

### 2.2. Hemostatic Parameters


FibrinogenPlatelets, platelet glycoproteins (GP) Ia, Ib, and IIIaMean platelet volume (MPV)Prothrombin time (PT), activated partial thromboplastin time (aPTT)Factor VIIIFactor V 1691 G-A, Factor V 4070 A-G, prothrombin 20210 G-A(PT 20210G-A), methylene tetrahydrofolate reductase (MTHFR) 677 C-T mutation


One major cause of SSNHL may be a sudden reduction of blood flow in the labyrinthine artery. Plasma viscosity and local regulatory mechanisms are factors responsible for the regulation of adrenergic receptors in smooth muscle, which, in turn, regulate blood flow through the labyrinthine arteries [17,18]. Glycoproteins (GP) Ia, Ib, and IIIa are involved in platelet adhesion and fibrinogen binding, and fibrinogen is ultimately involved in final clot formation. Hearing recovery in patients with SSNHL was significantly associated with low fibrinogen concentration (*p* = 0.029) and low GP IIIa receptor density (*p* = 0.037), suggesting a correlation between SSNHL and the mechanism of thrombosis in labyrinthine artery through interactions between platelet GP and fibrinogen [19].

Cochlear microcirculation has been associated with hearing. A study in guinea pigs reported that high serum fibrinogen is related to cochlear blood flow reduction, and that administration of a defibrinogenation drug enhances cochlear blood flow [20]. This was particularly effective in patients with profound hearing loss and high initial plasma fibrinogen levels [21]. A search of the PubMed and Scopus databases found 19 articles about SSNHL and fibrinogen. Although a meta-analysis showed no difference in fibrinogen concentrations between the control and SSNHL groups, serum fibrinogen concentrations were significantly lower in SSNHL patients who did recover than those who did not (*p* = 0.027). Moreover, high fibrinogen level was associated with poor prognosis, and was more strongly related to hearing severity or prognosis than to the cause of SSNHL. Nevertheless, the mechanism by which hyperfibrinogenemia affects the incidence of SSNHL, which involves multiple pathological phenomena, is unclear, although these findings suggest that an appropriate treatment method should be based on the cause of SSNHL [22].

Because impaired cochlear blood circulation is thought to be the main etiologic cause of SSNHL, genetic mutations that increase the risk of thrombosis may be biomarkers for SSNHL. Assessments of gene mutations by reverse transcription polymerase chain reaction (RT-PCR) showed that only the MTHFR 677 C-T mutation rate was significantly higher in patients with SSNHL than in control subjects (*p* = 0.03). In contrast, other mutations, such as the FV 1691 G-A mutation and PT 20210 G-A mutation, did not show statistically significant differences. This study suggested that a genetic predisposition to thrombosis may be a risk factor for SSNHL [23].

### 2.3. Inflammatory Parameters


Neutrophil/lymphocyte ratio (NLR)Platelet/lymphocyte ratio (PLR), lymphocyte countMonocyte count, monocyte/lymphocyte ratio (MLR)CRP (C-reactive protein)/albumin ratio, hs (high sensitivity)-CRPESR (erythrocyte sedimentation rate)Procalcitonin


Neutrophils, which have a thrombogenic profile, are both risk factors for and prognostic of stroke and myocardial infarction. Because the onset mode of SSNHL is the same as that of heart and brain infarction, the thrombogenic profile of neutrophils may be associated at least with severe SSNHL. Neutrophil count in patients with SSNHL was found to correlate positively with hearing level at the initial visit (*r* = 0.64, *p* = 0.00001) and negatively with recovery rate after one week (r = −0.63, *p* = 0.00003) and final recovery rate (*r* = −0.63, *p* = 0.00002). These findings suggested that neutrophil counts exceeding the reference range could be a clinical indicator of SSNHL severity and prognosis and may be related to the pathogenesis of idiopathic sudden hearing loss (ISHL) [24]. 

Another study comparing blood cell populations in SSNHL patients and controls showed that white blood cell (WBC), neutrophil, monocyte, lymphocyte, and platelet counts, as well as neutrophil/lymphocyte ratio (NLR) and platelet/lymphocyte ratio (PLR), were significantly higher in the SSNHL than in the control group. In contrast, WBC, absolute neutrophil, absolute monocyte, and platelet counts did not differ significantly between patients who did and did not recover from SSNHL. In addition, age, sex, body mass index (BMI), and concentrations of blood urea nitrogen (BUN), creatinine (Cr), alanine aminotransferase (ALT), and aspartate aminotransferase (AST) did not differ significantly between patients who did and did not recover from SSNHL. However, mean lymphocyte count was lower (*p* = 0.014), and NLR and PLR significantly higher *(p* < 0.001 each), in the recovered group. These findings suggested that NLR and PLR, especially the former, are useful prognostic markers in patients with SSNHL [25].

Factors thought to be related to SSNHL include WBC, monocyte, and lymphocyte counts, monocyte/lymphocyte ratio (MLR), NLR, and PLR, procalcitonin, interleukin 6 (IL-6), C-reactive protein (CRP), high-sensitivity-CRP (hs-CRP), and tumor necrosis factor-α (TNF-α) concentrations, and erythrocyte sedimentation rate (ESR). Chronic inflammation, one of the many causative factors of SSNHL, increased the risk of microvascular damage and ischemia, which can lead to atherogenesis [26]. Blood is supplied to the cochlea by a single vessel, the labyrinthine artery, not by multiple blood vessels, and cochlear hair cells are extremely vulnerable to hypoxia resulting from high oxygen consumption, making the cochlea sensitive to alterations in blood circulation. Thus, the pathophysiology of SSNHL is very closely associated with the cochlear vasculature. Moreover, circulating inflammatory molecules exert harmful effects on tissues of the cochlear vasculature.

Inflammation can both induce SSNHL and affect its prognosis. NLR is the most researched factor, with 10 papers to date assessing its relationship to the diagnosis of SSNHL and nine assessing its prognostic ability compared with other inflammation-related factors, including WBC count, IL-6, CRP, and TNF-α concentrations, NLR, and PLR. A meta-analysis of these studies, involving 1029 patients with SSNHL and 1020 healthy control subjects, found that patients with SSNHL had significantly higher NLR than control subjects (standardized mean deviation (SMD) = 1.65, 95% confidence interval (CI) = 1.20–2.09, *p* < 0.001). Moreover, NLR was significantly higher in patients who did not recover than those who did (SMD = 1.27, 95% CI: 0.62–1.92, *p* < 0.001). Taken together, these findings indicate that NLR may be a useful biomarker for the onset and prognosis of SSNHL [27,28].

No study to date has proposed the most appropriate NLR cut-off in normal people, but a recent study suggested that the normal range was 0.78–3.53 [29]. Individual markers associated with inflammation in patients with SSNHL include higher WBC, neutrophil, monocyte, and lymphocyte counts, and composite markers associated with inflammation in these patients include higher NLR and PLR. Except for lymphocyte count, no single marker of inflammation differed significantly in patients who did and did not recover after treatment. Due to their vulnerability to various factors, single markers of inflammation are regarded as less credible. In contrast, composite markers such as NLR and PLR are relatively stable and can be measured easily and cost-effectively [30]. 

NLR was found to correlate linearly with recovery in patients with SSNHL. At all frequencies, patients with low (<6.661) NLR showed better hearing recovery than patients with high (≥6.661) NLR, with more significant results observed at frequencies of 500, 1000, 2000, 3000, and 4000 Hz. Because recovery is reduced as NLR is increased, NLR could be a significant prognostic marker in patients with SSNHL [31].

SSNHL can occur at all ages, but it is rare in children. Because NLR and PLR were found to be associated with SSNHL in adults [32,33], these associations were also investigated in children. NLR differed significantly between children with SSNHL and controls and between children with SSNHL who did and did not recover after treatment. In contrast, PLR did not differ significantly between these groups. Therefore, NLR is considered a significant prognostic factor in children [34].

Platelets are involved in the pathophysiology of inflammation, coagulation, thrombosis, and vessel atherosclerosis [35]. Mean platelet volume (MPV) has been reported to be an indicator of platelet activation [36]. Large platelets produce proteins that are involved in hemostatic, vasomotor, and proinflammatory activities. Elevated MPV is a risk factor for thrombovascular disease, whereas low lymphocytic count is associated with viral inflammation. PLR is used to evaluate the degree of systemic inflammation and endothelial damage to the peripheral vascular system, with increased PLR being associated with increased platelet adhesion to recently damaged blood vessels. MPV and PLR are both recognized as independent prognostic factors of cardiovascular disease [37]. SSNHL is thought to be related to inflammation, ischemia, thrombosis, and embolism. A study assessing the differences between audiogram and platelet-related factors in SSNHL found that lymphocyte count, MPV, and PLR were associated with SSNHL but not with audiographic results. However, low lymphocyte count and elevated MPV and PLR were associated with prognosis in patients with high-frequency SSNHL, and higher MPV level was associated with poor prognosis in patients with all-frequency SSNHL [38].

CRP is a ring-shaped pentamer protein found in human plasma that is increased under conditions of inflammation. CRP is classified as an acute phase protein, with its blood concentration increased following the secretion of IL-6 by macrophages and T cells. CRP concentration may be indicative of infection and the efficacy of treatment [39]. A study that analyzed hs-CRP and procalcitonin found that only procalcitonin concentration was significantly higher in patients with SSNHL than in control subjects, suggesting that procalcitonin may be indicative of inflammatory etiology in patients with SSNHL [40]

The concentration of albumin (Alb), a basic component of cells with a molecular weight of 65–70 kDa, may also be prognostic of short- and/or long-term inflammation. Alb concentration may be reduced under conditions of acute and chronic inflammation and malnutrition [41]. CRP/Alb ratio was shown to be a prognostic factor in various inflammatory conditions, including in tumors accompanied by inflammation. Moreover, the CRP/Alb ratio was found to be significantly higher in patients with SSNHL than in control subjects (0.95 ± 0.47 vs. 0.74 ± 0.13, *p* = 0.009). This ratio did not differ significantly between patients who did and did not respond to treatment but was lower in responders. Taken together, these findings suggest that the CRP/Alb ratio may be a prognostic marker of SSNHL [42].

### 2.4. Immunologic Parameters


Toll-like receptors (TLRs) 2, 3, 4, 7, 8, and 9Myeloid differentiation primary response 88 (MyD88), Interferon regulatory factors (IRF) 3 and 7, Interleukin 1 receptor associated kinase 1 (IRAK1), REL-associated protein involved in NF-κB (RELA), TANK binding kinase (TBK1), TNF receptor-associated factors (TRAF) 3 and 6,TNF-α, sCD40 (soluble CD40), sCD40L (sCD40 ligand)IL-6, 10, 12Natural killer cell activity (NKCA)


The ability of all of these immune response-associated factors to act as developmental and prognostic markers of SSNHL has been analyzed. The blood–labyrinthine barrier is thought to separate the inner ear from systemic cellular and humoral immunity. Recently, however, local immune responses were observed in the inner ear due to the increased movement of immunocompetent cells in the bloodstream of the inner ear [43]. Thus, the inner ear is no longer regarded as an immunologically privileged site, suggesting that an immune system-mediated mechanism may be involved in the etiopathogenesis of SSNHL [44].

Serum concentrations of TNF-α, IL-10, and IL-12 were found to be similar in SSNHL patients and control subjects. However, TNF-α levels in non-responders to treatment were higher after than before treatment. These findings suggested that IL-10 and IL-12 do not play important roles in SSNHL, whereas TNF-α is associated with the pathophysiology of ISSHL, suggesting that TNF-α receptor blockers may be effective in these patients [45]. However, another study found that TNF-α concentration was higher in SSNHL patients than in control subjects, with dendritic cells playing a role in SSNHL [46].

CD40, a member of the TNF-α superfamily, is an immune regulatory factor involved in the secretion of pro-inflammatory chemokines and the recruitment and adhesion of leukocytes. Activated CD40 ligand (sCD40L) is upregulated in patients with lupus [47]. The serum concentrations of sCD40 and sCD40L were found to be significantly higher, whereas T lymphocyte counts tended to be lower, in patients with SSNHL than in control subjects, suggesting that the extravasion process of pro-adhesive and proinflammatory lymphocytes may be related to the progression of SSNHL [48].

SSNHL occurs following a common cold or mumps or measles virus infection, with some patients with SSNHL being positive for antibodies against adenovirus, cytomegalovirus, Epstein-Barr virus, herpes simplex virus, influenza virus, mumps virus, respiratory virus, and varicella-zoster virus [49,50]. These findings suggested that viral infection may induce SSNHL and that the innate immune response may be involved in its pathophysiology. The innate immune response is the body’s primary defense system, which reacts immediately to a pathogen without remembering specific pathogens. Pattern-recognition receptors (PRRs) are important in innate immunity, as they recognize pathogen-associated molecular patterns (PAMP) in pathogenic microbes, including bacteria, viruses, parasites, fungi, and protozoans. This recognition induces the mobilization of cytokines, chemokines, and interferons. PRRs involved in maintaining a first line of defense in hosts include C-type lectin receptors (CLRs), nucleotide-binding oligomerization domain-like receptors (NLRs), retinoic acid-inducible gene 1 (RIG-I), and TLRs. The levels of expression of TLR2, TLR3, TLR4, TLR7, TLR8, and TLR9 were found to be higher in SSNHL patients than in healthy controls (*p* < 0.05 each), with TLR2 gene expression being especially high in patients with profound hearing loss > 90 dB (*p* < 0.05). These findings indicate that TLRs, which are involved in innate immunity, play important roles in SSNHL, with TLR2 being specifically related to the severity of hearing loss [51].

Other types of systemic stress may trigger the abnormal activation of NF-κB (nuclear factor kappa-light-chain-enhancer of activated B cells) in unilateral cochlear lateral walls, inducing SSNHL. Generally, NF-κB plays pivotal roles in immune and inflammatory responses [52]. To verify the systemic stress response theory of SSNHL, various biomarkers associated with systemic stress and cochlear inflammation, such as NKCA, WBC counts, IL-6, TNF, NF-κB, and high-sensitivity CRP (hCRP), were investigated in SSNHL patients [53]. Neutrophil counts above the proposed reference range showed associations with severe hearing loss and poor prognosis, along with low NKCA and IL-6 concentrations. This correlation, however, was not evident in healthy control subjects. Taken together, these results suggest that increased neutrophil counts are a marker of poor prognosis, and that decreased NKCA, an acute increase in neutrophil count, and increased IL-6 may activate NF-κB in the cochlea, inducing severe SSNHL. Additional epidemiologic investigations are needed to determine whether a high-stress environment increases the risk of severe SSNHL [24].

### 2.5. Oxidative Factors 


Total oxidant status (TOS), total antioxidant status (TAS), paraoxonase (PON)thiol/disulfide ratiooxidative stress index (OSI)anti heat shock protein 70 (anti-HSP 70)


Reactive oxygen species (ROS) have been observed in the inner ear perilymph of patients with profound SSNHL during cochlear implant surgery, and damage to and apoptosis of inner ear hair cells were reported to be due to ROS-mediated injury [54]. Noise, drugs, heredity, aging, and Meniere’s syndrome are associated with cochlear damage, increasing ROS production, and suggesting that oxidative stress may be the main cause of SSNHL [55]. Although total oxidant status (TOS) and oxidative index were higher in patients with SSNHL than in normal controls, there were no between-group differences in total antioxidant status (TAS), paraoxonase (PON), native thiol, or total thiol levels, and no association between oxidative markers and the severity of hearing loss. Although this was a preliminary study, hypoxia was found to induce endothelial dysfunction of the inner ear, further damaging its microcirculation. Taken together, these findings suggest that oxidative stress is involved in the etiopathogenesis of ISSNHL [56].

In contrast, another study found that PON was associated with prognosis in patients with SSNHL. In a study analyzing PON and anti-heat shock protein 70 (anti-HSP 70), PON was found to be higher in the group of SSNHL patients than in the control group both before and after treatment (*p* < 0.05). In addition, anti-HSP 70 concentration was higher before and after treatment in patients who did than did not recover than in those who did recover (*p* < 0.05). These results suggested that PON is a factor that can be used to evaluate SSNHL, whereas anti-HSP 70 is prognostic in these patients [57].

### 2.6. Other Factors


Prestin, prestin antibodiesCoenzyme Q10Red blood cell distribution width (RDW)


Other factors, including prestin, anti-prestin autoantibodies, and coenzyme Q10, may be prognostic in patients with SSNHL. Prestin, an outer hair cell (OHC)-specific protein, has been reported to be a biochemical marker for early diagnosis of SNHL loss in an animal model of noise-induced hearing loss and cisplatin ototoxicity [58,59]. Prestin concentration in serum, as measured by ELISA, was reported to be significantly higher in ISSHL patients than in normal controls (*p* < 0.001). However, prestin concentrations were lower after than before treatment in 60% ISSHL patients who responded to treatment, although the difference was not statistically significant. Another study, in which anti-prestin autoantibodies were analyzed by ELISA in plasma of SSNHL patients, found no statistically significant differences [60,61]. 

Univariate analysis showed that ISSNHL was associated with high total cholesterol (*p* < 0.05), high LDL (*p* = 0.021), and low CoQ (*p* < 0.05) concentrations. In contrast, multivariate analysis showed that only high total cholesterol and low CoQ levels were independent risk factors for SSNHL. These results suggested a correlation between ISSNHL and low concentrations of the antioxidant CoQ in serum [62,63].

Red blood cell distribution width (RDW), a parameter used to classify anemia, was recently reported to be associated with inflammation and microcirculation disorders and with diseases such as coronary artery disease and rheumatoid arthritis [64,65]. RDW may also be a biomarker in patients with SSNHL, being significantly higher in SSNHL patients who did not recover than those who did recover after treatment (*p* = 0.031). Moreover, RDW and hearing recovery showed a significant negative correlation (*p* = 0.03, *r* = −0.24), suggesting that RDW may be a significant prognostic factor in patients with SSNHL [66].

The present study had limitations. First, the literature search of Embase, PubMed, and the Cochrane Library was conducted using only the terms ‘Sudden sensorineural hearing loss, biomarker, and prognostic’. Publications related to the topic may have been missed because they were not identified in searches using the above key words. Second, we reviewed biomarkers, but could not provide reference values or cut-off values. The statistical analyses in the reviewed papers were conducted based on the mean values of specific factors in the patient and control groups. In most of the papers that identified prognostic biomarkers, the statistical analyses were performed based on the mean value of specific factors in the recovered and non-recovered groups, and cut-off values were not suggested. We expect that cut-off values will be suggested in future biomarker research.

## 3. Conclusions

Several biomarkers have been associated with outcomes in patients with SSNHL. Good prognosis in patients with SSNHL was associated with low NLR and NHR, low monocyte counts, and low fibrinogen, GP IIIa, and TNF-α concentrations. However, because studies on biomarkers of SSNHL are still ongoing, no firm conclusions can be drawn. Table 1 below shows a summary of biomarkers for idiopathic sudden sensorineural hearing loss.

## Figures and Tables

**Table 1 ijms-21-07248-t001:** Summary of biomarkers for idiopathic sudden sensorineural hearing loss.

Author (Year)	Study Design	*n*	Age (Years)	Biomarker	Outcome	Conclusion
Öçal et al. (2020) [42]	Retrospective	40 patients, 45 controls	44.1 ± 14.2, 42.2 ± 13.8	CRP/Alb ratio, NLR	Mean CRP/Alb ratio was 0.95 ± 0.47 in the patient group and 0.74 ± 0.13 in the control group (*p* = 0.009). Mean CRP/Alb ratio and mean NLR were not significantly related.	CRP/Alb ratio was significantly higher in patients with SHL than in the control group.
Kang et al. (2020) [31]	Retrospective, case-control study	137 patients	≥ 45	Hb, WBC, neutrophils, lymphocytes, monocytes, NLR, platelets, PLR, glucose, HbA1c, BUN, Cr, AST, ALT, TC, TGs	NLR showed a linear correlation with hearing recovery in patients with SSNHL, with hearing gain (dB) = 56.698 − 3.718 × NLR (*r*^2^ = 0.451, *p* = 0.001). Hearing recovery at all frequencies was numerically higher in patients with low (< 6.661) than higher (≥ 6.661) NLR at all frequencies and was significantly higher at 500, 1000, 2000, 3000, and 4000 Hz. Hearing thresholds at 250, 500, 1000, 2000, 3000, and 4000 Hz in the low NLR group were significantly lower after treatment.	High NLR is associated with poor outcome of patients with SSNHL.
Zhang Weng et al. (2019) [14]	Retrospective	33 successive bilateral, 215 unilateral	48.67 ± 15.36, 42.71 ± 13.58	NLR, PLR, MLR, HDL, LDL, TC, TGs, Fibrinogen, PT, aPTT	NLR, MLR, and PLR in the successive bilateral SSNHL group were significantly higher (NLR: 5.72 ± 2.23 vs. 4.45 ± 2.82, *p* = 0.01; MLR: 0.25 ± 0.15 vs. 0.17 ± 0.11, *p* < 0.01; PLR: 190.70 ± 69.79 vs. 148.18 ± 65.67; *p* < 0.01); LDL level was significantly higher; HDL level was significantly lower (LDL: 3.79 ± 0.53 vs. 3.49 ± 0.74; HDL: 1.33 ± 0.32 vs. 1.44 ± 0.26; *p* < 0.05 for both); fibrinogen was significantly higher (4.03 ± 0.47 vs. 3.70 ± 0.65; *p* < 0.01).	NLR, PLR, MLR, LDL, fibrinogen and diabetes showed positive correlations with successive bilateral SSNHL. However, HDL was negatively correlated with successive bilateral SSNHL.
Sun and Xuan et al. (2019) [61]	Clinical study	14 patients, 24 controls	57.9 (15.4), 54.0 (9.9)	Prestin	Mean prestin concentration was 840.24 ± 496.22 pg/mL in the control group and 955.98 ± 2501.48 pg/mL in the patient group (*p* < 0.001).	Prestin may be a diagnostic rather than a prognostic biomarker in SSNHL.
Kaneva et al. (2019) [16]	Case-control study	27 patients, 24 controls	39.7 (27–51), 32.3 (25–47)	TC, TG, HDL, LDL, apoA-1, apoB, apoE, AIP, ATH index	No significant differences in TC, TGs and HDL-C between ISSNHL patients and the control group. Higher values of the apoB/apoA-I ratio, AIP and ATH index in patients with SSNHL indicated increased atherogenicity of the lipid profile.	High ATH index value is associated with the occurrence of SSNHL.
Tovi et al (2019) [60]	Retrospective	63 patients	47 ± 16 (18–77)	Prestin antibodies, C3, C4, ANA, ENA	No statistically significant association was found between prestin autoantibodies and audiologic parameters.	Anti-prestin antibodies are not a diagnostic biomarker for ISSNHL.
Qiao et al. (2019) [33]	Retrospective	60 patients 60 controls	45.62 ± 13.16, 49.62 ± 10.66	WBC, neutrophils, platelets, lymphocytes, NLR, PLR	Mean NLR and PLR were significantly higher in patients than controls and were significantly higher in those who did not than those who did respond to treatment.	Low NLR and PLR are associated with good outcome of SSNHL.
Yoon and Kim et al. (2019) [46]	Prospective	24 patients, 24 controls	43.12, 46.91	TNF-α, IL-10, 12, IFN-γ, mononuclear cells, clusters of differentiation 11c and 86	Mean percent monocytes (26.36% ± 4.3% vs. 14.32% ± 2.3%) and mean TNF-α level (15.8 ± 9.3 pg/mL vs. 12.4 ± 8.7 pg/mL) were significantly higher in the SSNHL than in the control group. Mean IFN-γ and IL-12 levels were significantly lower in the SSNHL group than the control group.	Increases in TNF-α level and monocyte population, and decreases in IFN-γ and IL-12 levels, might have critical roles in SSNHL.
Chen et al. (2018) [27]	Systematic review, meta-analysis	1029 patients, 1020 controls		NLR	NLR levels were higher than in the patient than in the control group (SMD = 1.65, 95% CI = 1.20–2.09, *p* < 0.001). NLR value was much higher in non-recovered patients than in recovered patients (SMD = 1.27, 95% CI: 0.62–1.92, *p* < 0.001).	Low NLR is associated with good outcome of SSNHL.
Oya et al. (2018) [22]	Meta-analysis	1577 patients		Fibrinogen	Fibronectin concentration did not differ significantly between SSNHL patients and controls. Serum fibrinogen level of the recovery group was significantly lower than that of the nonrecovery group (*p* = 0.027).	High fibrinogen level is associated with poor outcome of SSNHL.
Göde et al. (2018) [40]	Retrospective	23 patients 19 controls	47.91 ± 15.73, 35.16 ± 15.67	Procalcitonin, hs-CRP	Procalcitonin levels were significantly higher in patients than in the control group (*p* = 0.018).	High procalcitonin level is associated with SSNHL.
Sun et al. (2017) [38]	Retrospective	129 patients, 31 controls	44.68 ± 9.11, 43.00 ± 16.44, 43.44 ± 12.81, 43.69 ± 19.06, 51.06 ± 10.01	Platelets, lymphocytes, MPV, PLR	There was significant difference between each study group and the control group in terms of lymphocyte count (all *p* < 0.01). Compared to the control group, MPVs of AF-SSNHL and TD-SSNHL patients were significantly higher (*p* < 0.01) and PLRs were significantly higher (*p* = 0.03, *p* < 0.01, *p* < 0.01, and *p* < 0.01 for LF-, HF-, AF-, and TD-SSNHL, respectively). Lymphocytes, MPV, and PLR of the HF-SSNHL subgroup (all *p* < 0.01) and MPV of the AF-SSNHL subgroup (*p* = 0.04) differed significantly from those of the other subgroups.	Low MPV is associated with good outcome of AF-SSNHL. High lymphocytes and PLR, and low PMV, are associated with good outcome of HF-SSNHL.
Bulgurcu et al. (2017) [34]	Retrospective	21 patients, 24 controls	13.7 ± 3.2, 14.8 ± 2.9	Neutrophils, lymphocytes, platelets, NLR, PLR	Neutrophils, lymphocytes and NLR differed significantly between the patient and control groups (*p* = 0.017, *p* = 0.039 and *p* = 0.016, retrospectively).	Low NLR is an important marker of good prognosis of ISSNHL in pediatric patients.
Fasano et al. (2017) [15]	Retrospective, case-control study	131 patients, 77 controls	54, 52.5	Glucose, HbA1c, Uric acid, ALT, AST, Cr, CPK, TSH, CRP, Factor VIII, PT, TC, aPTT, homocysteine, fibrinogen, mutation prothrombin, LDL, mutation factor V, LP(a), HDL, TGs,	Blood glucose, HbA1C, Lp(a), and factor VIII concentrations were significantly higher in patients than controls (*p* < 0.05). Furthermore, blood glucose, HbA1C, uric acid, factor VIII, and homocysteine concentrations were significantly associated with severity of SSNHL.	The severity of SSNHL is associated with high blood glucose, HbA1c, uric acid, factor VIII, and homocysteine.
Gul et al. (2017) [56]	Retrospective	50 patients, 50 controls	43.98 ± 11.69, 43.5 ± 9.19	TOS, TAS, PON, OSI calculation, thiol/disulfide ratio	TOS, OSI, disulfide, disulfide/native thiol, disulfide/total thiol, and native thiol/total thiol levels differed significantly between the study and control groups (*p* = 0.008, *p* = 0.018, *p* = 0.001, *p* = 0.006, *p* = 0.002, *p* = 0.002, respectively).	Increases in TOS, OSI and native thiol/total thiol, and decreases in disulfide, disulfide/native thiol and disulfide/total thiol, are associated with SSNHL.
Koçak et al. (2016) [13]	Retrospective, case-control clinical trial.	45 patients, 47 controls	31.1 (+7.4), 32.4 (+8.1)	Monocytes, HDL, MHR	MHRs did not differ significantly between patients and controls. (*p* = 0.574). However, the MHR was significantly higher in non-responders compared with responders (*p* = 0.005)	Low monocyte and MHR are associated with good outcome of SSNHL.
Oya et al. (2016) [21]	Prospective	61 defibrinogenation, 64 steroid	59.2 ± 14.9, 57.3 ± 16.5	PT-INR, aPTT, fibrinogen, PIC, SFMC, AT-III, PMG, α2PI, TAT	In patients who recovered completely, serum fibrinogen level before treatment was significantly higher in the defibrinogenation group than in the corticosteroid group.	Defibrinogenation therapy is more appropriate than corticosteroid therapy for profound hearing loss with high initial serum fibrinogen concentration.
Nonoyama et al. (2016) [66]	Retrospective	89 patients	54.2 ± 17.5	RBC, Hb, RDW, WBC, platelets, glucose, BUN, Cr, AST, ALT, NLR, MPV	Mean RDW was significantly higher in the non-recovered group (13.2% ± 1.0% compared with 12.7% ± 0.7% in the recovered group, *p* = 0.031) in a binary logistic regression model, RDW was associated with recovery from ISSNHL (odds ratio = 2.33, 95% confidence interval = 1.20–4.51, *p* = 0.012).	Higher RDW is associated with poor outcome of SSNHL.
Kum et al. (2015) [28]	Retrospective cross-sectional historical cohort	59 patients 59 controls	46.10 ± 11.91, 42.84 ± 11.85	NLR, MPV, platelets, WBC, neutrophils, lymphocytes	NLR levels were much higher in patients diagnosed with sudden hearing loss compared to the control group. Similarly, mean NLR was higher in non-recovered versus recovered patients (*p* = 0.001).	Low LNR is associated with good outcome of SSNHL.
Yang et al. (2015) [51]	Retrospective	36 patients, 71 controls	50.94 ± 18.62 50.25 ± 13.27	TLR2,3,4,7,8,9, MyD88, IRAK1, TRAF3,6, TBK1, IRF3,7, RELA	Expression levels of TLR2, 3, 4, 7, 8, and 9 genes were significantly higher in SSNHL patients than in normal controls (*p* < 0.05). Higher expression of the TLR2 gene was found in patients with profound hearing loss compared with those with less severe hearing loss (*p* < 0.05).	Higher expression of the TLR2 gene is associated with severity of SSNHL.
Seo et al. (2014) [25]	Retrospective	348 patients, 53 controls	48.19 ± 15.22, 48.22 ± 11.60	WBC, neutrophils, lymphocytes, monocytes, NLR, platelets, PLR, glucose, BUN, Cr, AST, ALT	Mean NLR and PLR were significantly higher in the patient than in the control group (*p* < 0.001). NLR was significantly higher in the non-recovered than in the recovered group (5.98 ± 4.22 vs. 3.50 ± 3.3, *p* < 0.001)	Low LNR is associated with good outcome of ISSNHL.
Weiss et al. (2014) [19]	Retrospective	127 patients, 81 controls	53.3 ± 17.1, 49.9 ± 12.6	Platelet glycoprotein (GpIa, GpIb, and GpIIIa) receptor densities, fibrinogen	Lower fibrinogen levels (*p* = 0.029) and lower GpIIIa receptor density (*p* = 0.037) were associated with hearing recovery.	Low fibrinogen and GpIIIa receptor density are associated with good outcome of SSNHL
Düzer et al. (2014) [57]	Retrospective	25 patients, 25 controls	39.48 (16–65), 34.16 (21–59)	Anti-HSP 70, PON	Pre- and post-treatment serum PON levels were significantly higher in the patient group than in the control group (*p* < 0.05). In patients with complete or partial recovery, pre- and post-treatment serum anti-HSP70 levels were significantly higher than in controls (*p* < 0.05).	High PON was associated with SSNHL. High anti-HSP 70 was associated with good outcome of SSNHL.
Demirhan et al. (2013) [45]	Prospective clinical trial	23 patients, 20 controls	52 (35–67) (age- and sex-matched)	TNF-α, IL-10,12	There were no significant differences between pre- and post-treatment values of TNF-α in treatment responders (*p* > 0.05). In treatment non-responders, post-treatment TNF-α was elevated compared to the pre-treatment value (*p* < 0.05).	High TNF-α is associated with poor outcome of SSNHL.
Masuda et al. (2012) [24]	Individual cohort study	43 patients, 10 controls	57 ± 15, 52 ± 15	Neutrophils, lymphocytes, monocytes, NKCA, IL-6, TNF, hs-CRP	Neutrophil counts above the reference range of a facility will be a useful indicator for poor prognosis of ISSHL.	High neutrophil count is associated with poor outcome of ISSHL.
Cadoni et al. (2010) [63]	Case-control study	43 patients, 43 controls	50 ± 14, 43 ± 11	TC, LDL, coenzyme Q10, platelet, prothrombin time, fibrinogen, ESR, CRP, serum gamma globulin level	On univariate logistic regression analysis, significant associations were found between SSNHL and high serum total cholesterol level (*p* < 0.001), high LDL level (*p* = 0.024), and low coenzyme Q10 level (*p* < 0.001). On multivariate logistic regression analysis, statistically significant associations were found between low nervonic acid (*p* = 0.005) (unsaturated), low coenzyme Q10 (*p* = 0.002), and high total cholesterol (*p* = 0.015) serum level and high risk of SSNHL.	Low nervonic acid, low coenzyme Q10, and high TC are associated with SSNHL.
Kassner et al. (2011) [48]	Retrospective	12 patients, 12 controls	45.0 ± 3.2, 45.4 ± 4.1	TC, HDL, LDL, TGs, lymphocytes, monocytes, neutrophils, eosinophils, basophils, leukocytes, CRP, sCD40, sCD40L, TNF-α	Acute SHL patients had high levels of sCD40 and sCD40L and a significantly decreased percentage (36%) of lymphocytess, especially T lymphocytes (28%). Proinflammatory CD40, TNF-α, cyclooxygenase-2, or CD38-positive T or B lymphocytes were significantly increased.	Low T lymphocytes, high sCD40, and high sCD40L are associated with SSNHL.
Yildiz et al. (2008) [23]	Retrospective	53 patients 80 controls	4–63, 18–60	FV 1691 G-A, PT 20210 G-A, MTHFR 677 C-T, FV 4070 A-G, EPCR gene 23-bp insertion, PAI-1 4G/5G mutation	The frequency of MTHFR 677 C-T mutation was significantly higher in the patient group than in the control group (*p* = 0.03).	MTHFR gene 677 C-T mutation is associated with SSNHL.

Abbreviations: CRP/Alb ratio: C-reactive protein/albumin ratio; NLR: neutrophil to lymphocyte ratio; SHL: sudden hearing loss; Hb: hemoglobin; WBC: white blood cell; PLR: platelet to lymphocyte ratio; TC: total cholesterol; TGs: triglycerides; HbA1c: hemoglobin A1c; BUN: blood urea nitrogen; Cr: Creatinine; AST: aspartate transaminase; ALT: alanine transaminase; ISSNHL: idiopathic sudden sensorineural hearing loss; MLR: monocyte lymphocyte ratio; HDL: high density lipoproteins; LDL: low density lipoproteins; PT: prothrombin time; aPTT: activated partial thromboplastin time; SSNHL: sudden sensorineural hearing loss; apoA-1, B, E: apolipoproteinA-1, B, E; AIP: atherogenic index of plasma; ATH index: atherogenic index; ANA: anti-nuclear antibodies; ENA: anti-extractable nuclear antigen; TNF-α: tumor necrosis factor- α; IL-10, 12: interleukin-10, 12; IFN-γ: interferon- γ; hs-CRP: high-sensitivity-c-reactive protein; TOS: total oxidant status; TAS: total antioxidant status; PON: paraoxonase; OSI: oxidative stress index; MPV: mean platelet volume; AF-SSNHL: all-frequency SSNHL; TD-SSNHL: total-deafness SSNHL; HF-SSNHL: high-frequency SSNH; LF-SSNHL: low-frequency SSNHL; CPK: creatine phosphokinase; Lp(a): lipoprotein a; TSH: thyroid-stimulating hormone; MHR: monocyte to HDL ratio; PIC: plasmin-α2 plasmin inhibitor complex; SFMC: soluble fibrin-monomer complex; AT-III: antithrombin-III; PMG: plasminogen; α2PI: α2-plasmin inhibitor; TAT: thrombin-antithrombin III complex; RBC: red blood cell; RDW: red cell distribution width; TLR: Toll-like receptor; MyD88: myeloid differentiation primary response 88; IRAK1: interleukin-1 receptor-associated kinase; TRAF3, 6: tumor necrosis factor receptor-associated factor; TBK1: TANK-binding kinase 1; IRF3, 7: interferon regulatory factor; RELA: v-rel avian reticuloendotheliosis viral oncogene homolog A; Anti-HSP 70: anti-heat shock protein 70; PON: paraoxonase; NKCA: natural killer cell activity; ESR: erythrocyte sedimentation rate; sCD40: soluble CD40; sCD40L: sCD40 ligand; FV 1691 G-A: factor V 1691 G-A; PT 20210 G-A: prothrombin 20210 G-A; MTHFR 677 C-T: methylene tetrahydrofolate reductase 677 C-T; FV 4070 A-G: factor V 4070 A-G; EPCR: endothelial cell protein C receptor; PAI-1: plasminogen activator inhibitor-1.

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
