# Peer review of "Biomarkers Suggesting Favorable Prognostic Outcomes in Sudden Sensorineural Hearing Loss"

_ijms, 2020, doi:10.3390/ijms21197248_

Round 1
Reviewer 1 Report
The authors performed a review about biomarkers in sudden sensorineural hearing loss. It is an interesting topic on which several reviews and meta-analyses have already been executed. The main lack of the current manuscript is the focus on biomarkers, which is not distinguished from risk factors and has led to exclusion of numerous relevant publications.
Specific comments:
- abstract (line 18-19): 'the causes of SSNHL have not yet been identified', I would change this into 'etiological examinations in SSNHL do not always reveal a cause'.
- abstract (line 23): by controversially focusing on the term 'biomarker' (see also comment 5), you excluded publications relevant to your discussion, eg. PMID 29684383, PMID 25866869 and PMID 30232233 amongst others (which discuss the same but do not include the term 'biomarker')
- abstract (line 25): 'the remaining 28 articles were systematically reviewed', I would suggest to rephrase this as it suggests you performed a systematic review. Based on the provided data, you performed a review, although some strategies might fit a systematic review. For the latter, methods should be more elaborated, as well as adherence to certain guidelines (e.g. PRISMA).
- introduction (line 64): I believe the manuscript should benefit from stating a clear aim. Currently, it is not clear why you performed this review. Is there recent progress, is the field inconsistent on some points,...?
- metabolic parameters (line 99): I think you should comment on the difference between risk factors and biomarkers, although some overlap is possible. To me, e.g. ATH is a risk factor rather than a biomarker. Table 1 includes every possible factor as a biomarker, but then you should state which factor belongs to which category of biomarker. Moreover, some factors do not adhere to your definition in the introduction, stating that it should indicate response to treatment amongst others.
- inflammatory parameters (line 177): 'SMD', please write each abbreviation in full the first time
- general: the described factors can be linked to one specific cause of SSNHL (e.g. vascular, inflammatory,...). I wonder if a classification based on this instead of the factor type might be more relevant and appealing to the reader.
- general: I believe the paper lacks a critical analysis of the mentioned factors. If a factor is suggested as a prognostic or diagnostic factor, reference values should be provided (which was tackled in Table 2). However, cut-off values largely vary among different studies, again questioning the value of the specific factor, which merits further discussion. Moreover, quality of the study has not been taken into account.
Author Response
The authors performed a review about biomarkers in sudden sensorineural hearing loss. It is an interesting topic on which several reviews and meta-analyses have already been executed. The main lack of the current manuscript is the focus on biomarkers, which is not distinguished from risk factors and has led to exclusion of numerous relevant publications.
Specific comments:
- Point 1: abstract (line 18-19): 'the causes of SSNHL have not yet been identified', I would change this into 'etiological examinations in SSNHL do not always reveal a cause'.
Response 1: Thank you for your comment and suggestion. We have corrected the sentence as per your suggestion.
- Point 2 : abstract (line 23): by controversially focusing on the term 'biomarker' (see also comment 5), you excluded publications relevant to your discussion, eg. PMID 29684383, PMID 25866869 and PMID 30232233 amongst others(which discuss the same but do not include the term 'biomarker')
Response 2: In this review article, only papers identified in a search using the words ‘sudden sensorineural hearing loss, biomarker, and prognostic’ were reviewed. The papers you mention were not identified in this search and therefore could not be included.
- Point 3 : abstract (line 25): 'the remaining 28 articles were systematically reviewed', I would suggest to rephrase this as it suggests you performed a systematic review. Based on the provided data, you performed a review, although some strategies might fit a systematic review. For the latter, methods should be more elaborated, as well as adherence to certain guidelines (e.g. PRISMA).
Response 3: The word systematically is misleading and has been corrected as follows: “The remaining 28 articles were reviewed”.
- Point 4 : introduction (line 64): I believe the manuscript should benefit from stating a clear aim. Currently, it is not clear why you performed this review. Is there recent progress, is the field inconsistent on some points,?
Response 4: Thank you for your comment. SSNHL is mostly idiopathic because its etiology is not clear, and recovery is difficult to predict. The purpose of the present study was to evaluate the course of SSNHL and to identify a prognostic biomarker that can be used to predict treatment outcome.
- Point 5 : metabolic parameters (line 99): I think you should comment on the difference between risk factors and biomarkers, although some overlap is possible. To me, e.g. ATH is a risk factor rather than a biomarker. Table 1 includes every possible factor as a biomarker, but then you should state which factor belongs to which category of biomarker. Moreover, some factors do not adhere to your definition in the introduction, stating that it should indicate response to treatment amongst others.
Response 5: Thank you for your comment and suggestions. We agree that, even if there are overlapping concepts, risk factors and biomarkers should be distinguished. For clarity, we changed the phrase “suggesting that ATH index is a biomarker for SSNHL” to “ATH index is a risk factor for SSNHL.”
- Point 6 : inflammatory parameters (line 177): 'SMD', please write each abbreviation in full the first time
Response 6: Thank you for your comment. At first mention the abbreviation SMD has been defined as “standardized mean deviation”.
- Point 7 : general: the described factors can be linked to one specific cause of SSNHL (e.g. vascular, inflammatory...). I wonder if a classification based on this instead of the factor type might be more relevant and appealing to the reader.
Response 7: Thank you for your comment. As you mention, classification based on the cause of SSNHL might be more interesting to readers. In this review article, however, the metabolic, hemostatic, inflammatory, immunologic, oxidative, etc. classification was used because biomarkers were analyzed through blood tests, so the categories of blood tests were applied. In the next study, we will consider classification by cause of SSNHL.
- Point 8: general: I believe the paper lacks a critical analysis of the mentioned factors. If a factor is suggested as a prognostic or diagnostic factor, reference values should be provided (which was tackled in Table 2). However, cut-off values largely vary among different studies, again questioning the value of the specific factor, which merits further discussion. Moreover, quality of the study has not been taken into account.
Response 8: Thank you for your comment. The lack of a cut-off value or reference range is a limitation of this study. In the reviewed papers, statistical analysis was conducted based on the mean values of specific factors in the patient and control groups. In most papers that identify prognostic biomarkers, statistical analysis was performed based on the mean value of specific factors in the recovered and non-recovered groups, and cut-off values were not suggested. We expect that cut-off values will be suggested as biomarker research progresses in the future.
Reviewer 2 Report
Doo et al. performed a review in the literature to explore the biomarkers of the prognostic outcomes in sudden sensorineural hearing loss (SSNHL). The authors divided the factors into six categories: metabolic, hemostatic, inflammatory, immunologic, oxidative, and others, and then analyzed the associations between these factors with the patient prognosis. The authors’ literature review demonstrated that low monocyte counts, low neutrophil/lymphocyte ratio (NLR) and monocyte/HDL cholesterol ratio (MHR), and low concentrations of fibrinogen, platelet glycoprotein (GP) IIIa, and TNF-α were associated with good prognosis. However, these factors alone could not completely determine the onset of and recovery from SSNHL.
Comments
1. In general, the review was well conducted and the manuscript was well organized. It is not surprising that there are no predominant factors that can determine the outcomes of SSNHL, because the pathogenetic mechanisms of SSNHL are rather complex and remain largely unclarified.
2. Table 1 seems irrelevant to the main theme of this article and can be removed.
Author Response
Doo et al. performed a review in the literature to explore the biomarkers of the prognostic outcomes in sudden sensorineural hearing loss (SSNHL). The authors divided the factors into six categories: metabolic, hemostatic, inflammatory, immunologic, oxidative, and others, and then analyzed the associations between these factors with the patient prognosis. The authors’ literature review demonstrated that low monocyte counts, low neutrophil/lymphocyte ratio (NLR) and monocyte/HDL cholesterol ratio (MHR), and low concentrations of fibrinogen, platelet glycoprotein (GP) IIIa, and TNF-α were associated with good prognosis. However, these factors alone could not completely determine the onset of and recovery from SSNHL.
Comments
Point 1 : In general, the review was well conducted and the manuscript was well organized. It is not surprising that there are no predominant factors that can determine the outcomes of SSNHL, because the pathogenetic mechanisms of SSNHL are rather complex and remain largely unclarified.
Response 1 : Thank you for your comment. We agree with your opinion.
Point 2 : Table 1 seems irrelevant to the main theme of this article and can be removed.
Response 2 : Thank you for your comment and suggestion. Table 1 has been deleted.
Reviewer 3 Report
This work is conscientiously done. All important biomarkers are investigated and commented. However, the benefit for future therapy of SSNHL isn't clear yet. Which biomarker should be analyzed in patients with SSNHL routinely?
Author Response
This work is conscientiously done. All important biomarkers are investigated and commented. However, the benefit for future therapy of SSNHL isn't clear yet. Which biomarker should be analyzed in patients with SSNHL routinely?
Response: Thank you for your opinion. We believe that NLR (neutrophil-to-lymphocyte ratio) analysis is appropriate for SSNHL. We perform this test routinely in all SSNHL patients. First, because NLR is the most-studied prognostic biomarker of SSNHL. Second, because NLR can be easily obtained from the CBC (complete blood count) and DC (WBC differential count), which are the basis for blood tests.
Round 2
Reviewer 1 Report
The manuscript seems to have improved after revision. My only remaining concern is the methodology of the review. A review should be comprehensive in answering the question(s) stated in the aims. I agree that you did not include certain publications as they did not mention your search terms. However, if other publications also prove relevant for the topic, one might consider to adapt the search string in order to be more comprehensive and to better advance the field. Of course, this would result in a drastic change/addition. If you want to stick to the current methodology, the (too) strict search string should be mentioned in a limitations paragraph, in which you could also comment on the lack of cut-off values.
One additional minor comment is the change of numbering in the latter references (check the reference numbering from reference 62 onwards).
Author Response
Point 1 : The manuscript seems to have improved after revision. My only remaining concern is the methodology of the review. A review should be comprehensive in answering the question(s) stated in the aims. I agree that you did not include certain publications as they did not mention your search terms. However, if other publications also prove relevant for the topic, one might consider to adapt the search string in order to be more comprehensive and to better advance the field. Of course, this would result in a drastic change/addition. If you want to stick to the current methodology, the (too) strict search string should be mentioned in a limitations paragraph, in which you could also comment on the lack of cut-off values.
REPLY: Thank you for your advice. We agree that it is a limitation of this paper that we did not comprehensively search for papers related to the topic, and that we also could not provide cut-off values for biomarkers. We have mentioned these limitations in the revised manuscript.
“The present study had limitations. First, the literature search of Embase, PubMed, and the Cochrane Library was conducted using only the terms ‘Sudden sensorineural hearing loss, biomarker and prognostic’. Publications related to the topic may have been missed because they were not identified in searches using the above key words. Second, we reviewed biomarkers, but could not provide reference values or cut-off values. The statistical analyses in the reviewed papers were conducted based on the mean values of specific factors in the patient and control groups. In most of the papers that identified prognostic biomarkers, the statistical analyses were performed based on the mean value of specific factors in the recovered and non-recovered groups, and cut-off values were not suggested. We expect that cut-off values will be suggested in future biomarker research.”
Point 2 : One additional minor comment is the change of numbering in the latter references (check the reference numbering from reference 62 onwards).
REPLY: Thank you for your comment. The reference number seems to have been changed when we removed Table 1 during the revision process. We have corrected it.